# Dietary Fatty Acids Contribute to Maintaining the Balance between Pro-Inflammatory and Anti-Inflammatory Responses during Pregnancy

**DOI:** 10.3390/nu15112432

**Published:** 2023-05-23

**Authors:** Valeria Policastro, Dario Righelli, Lucilla Ravà, Pamela Vernocchi, Marzia Bianchi, Cristina Vallone, Fabrizio Signore, Melania Manco

**Affiliations:** 1Department of Political Sciences, University of Naples Federico II, 80138 Naples, Italy; valeria.policastro@gmail.com; 2Istituto per le Applicazioni del Calcolo “Mauro Picone”, National Research Council, 80131 Naples, Italy; 3Department of Statistical Sciences, University of Padova, 35121 Padua, Italy; 4Clinical Epidemiology, Bambino Gesù Children’s Hospital, IRCCS, 00165 Rome, Italy; lucilla.rava@opbg.net; 5Unit of Human Microbiome, Multimodal Laboratory Medicine Research Area, Bambino Gesù Children’s Hospital, IRCCS, 00165 Rome, Italy; pamela.vernocchi@opbg.net; 6Research Unit of Molecular Genetics of Complex Phenotypes, Bambino Gesù Children’s Hospital, IRCCS, 00146 Rome, Italy; marzia.bianchi@opbg.net; 7Obstetrics and Gynecology Department, USL Roma1, San Filippo Neri Hospital, 00135 Rome, Italy; cristina.vallone@aslroma1.it; 8Obstetrics and Gynecology Department, USL Roma2, Sant ‘Eugenio Hospital, 00144 Rome, Italy; 9Research Area for Fetal, Neonatal and Cardiological Sciences, Bambino Gesù Children’s Hospital, IRCCS, 00165 Roma, Italy

**Keywords:** dietary fatty acids, dietary lipids, inflammation, insulin resistance, obesity, pregnancy

## Abstract

Background: During pregnancy, the balance between pro-inflammatory and anti-inflammatory responses is essential for ensuring healthy outcomes. Dietary Fatty acids may modulate inflammation. Methods: We investigated the association between dietary fatty acids as profiled on red blood cells membranes and a few pro- and anti-inflammatory cytokines, including the adipokines leptin and adiponectin at ~38 weeks in 250 healthy women. Results: We found a number of associations, including, but not limited to those of adiponectin with C22:3/C22:4 (coeff −1.44; *p* = 0.008), C18:1 c13/c14 (coeff 1.4; *p* = 0.02); endotoxin with C20:1 (coeff −0.9; *p* = 0.03), C22:0 (coeff −0.4; *p* = 0.05); MCP-1 with C16:0 (coeff 0.8; *p* = 0.04); and ICAM-1 with C14:0 (coeff −86.8; *p* = 0.045). Several cytokines including leptin were associated with maternal body weight (coeff 0.9; *p* = 2.31 × 10^−5^), smoking habits (i.e., ICAM-1 coeff 133.3; *p* = 0.09), or gestational diabetes (i.e., ICAM-1 coeff 688; *p* = 0.06). Conclusions: In a general cohort of pregnant women, the intake of fatty acids influenced the balance between pro- and anti-inflammatory molecules together with weight gain, smoking habits, and gestational diabetes.

## 1. Introduction

During pregnancy, the woman’s body undergoes significant immune changes to support fetal implant and development, including an increase in certain immune cells and alterations in immune function that involve both pro-inflammatory and anti-inflammatory responses, which work together to promote maternal–fetal tolerance and fetal growth. A controlled inflammatory response helps to facilitate implantation and early placental development, and is characterized by the recruitment of immune cells, such as natural killer cells and macrophages, to the site of implantation, as well as the production of pro-inflammatory cytokines [1]. However, excessive inflammation can be harmful to fetal development and may increase the risk of complications such as preterm birth, pre-eclampsia, and fetal growth restriction [2]. Therefore, the balance between pro-inflammatory and anti-inflammatory responses is critical for ensuring healthy pregnancy outcomes [1].

In parallel, the woman’s body also experiences metabolic changes to accommodate the increased metabolic demands of fetal growth and development [3]. In early pregnancy, insulin sensitivity tends to increase, which helps to promote nutrient delivery to the developing fetus. This increased sensitivity is thought to be driven by hormonal changes, including increased levels of estrogen and progesterone, as well as other factors. As pregnancy progresses, insulin sensitivity tends to decline, particularly in the later stages of pregnancy [3]. In some cases, the excessive decline in insulin sensitivity can lead to a range of complications for both the mother and the baby, including gestational diabetes mellitus (GDM), pre-eclampsia, preterm birth, and large-for-gestational-age infants [3].

Reduced insulin sensitivity is deemed as a condition of low-grade systemic inflammation and can contribute significantly to worsening the physiological inflammation that characterizes pregnancy.

Dietary lipids can have a significant impact on inflammation through directly influencing the immune system [4] while worsening the systemic insulin sensitivity [5,6].

Consuming excessive amounts of saturated (SFAs) [7] and trans fats (TFAs) [8], commonly found in foods such as fried foods, baked goods, and processed snacks, have been linked to increased inflammation and a range of chronic diseases, including obesity, type 2 diabetes (T2D), and cardiovascular disease [7]. In contrast, consuming more monounsaturated (MUFAs) and certain polyunsaturated fats (PUFAs) have been associated with a reduced inflammation and improved health outcomes [5]. Healthy fats can help to reduce the production of pro-inflammatory cytokines, such as tumor necrosis factor alpha (TNF-alpha) and interleukin-6 (IL-6), while also increasing the production of anti-inflammatory cytokines, such as interleukin-10 (IL-10) [5]. The aim of the present study was to investigate the association between the dietary lipids estimated based on the profiling of total phospholipid fatty acids (FAs) on the red blood cell (RBC) membranes and inflammatory molecules at the end of the pregnancy and, importantly, to rule out any mediating roles of obesity and associated hyperinsulinemia. The lipid profile on the RBC membranes reflected the quality of dietary fats in the previous 40 days. Cytokines were evaluated at ~38 weeks when their levels were expected to change favoring the labor onset.

For these purposes, we measured circulating levels of specific molecules, namely adipokines and cytokines, which play an important role in immune function, inflammation, and metabolism, and whose imbalance has been found to contribute to the development of pregnancy-related complications such as GDM, pre-eclampsia, preterm birth and fetal growth restriction. We selected for testing molecules reflecting different facets of inflammation, and our list of molecules included adiponectin, leptin, soluble cluster of differentiation 14 (sCD14), soluble tumor necrosis factor receptor II (sTNFR-II), IL-6, IL-8, IL-10, monocyte chemoattractant protein 1 (MCP-1), endotoxin, and human intercellular adhesion molecule 1 (ICAM-1).

## 2. Materials and Methods

### 2.1. Subjects and Study Design

The “Feeding Low-Grade Inflammation and Insulin Resistance of the Fetus” study [9,10], a population study of 1000 mother–infant pairs, investigated the association between the dietary intake of fats during pregnancy, and the child’s insulin resistance and low-grade inflammation at birth as primary aim. The profile of fats on the RBC membranes was used as an estimate of the maternal intake. We enrolled healthy pregnant women with ages between 18 and 45 years from February 2013 until June 2015 at the San Camillo Forlanini Hospital (SCH) in Rome. We followed up enrolled women starting from the 1st trimester of pregnancy up to childbirth, with monitoring of their lifestyles, blood testing, and ultrasonography as recommended by national guidelines [11]. To be included in the study, women had to be at week 7–10 of gestation, take folic acid supplements from week 7, present a singleton pregnancy, consume no alcohol, take no medications, have no systemic, chronic, or autoimmune disease, no previous diagnosis of GDM or miscarriage, no conception through ovulation induction or in vitro fertilization, and plan delivery at the SCH-Unit. For this study, we only included the data of women with a complete data set of cytokines for our analysis.

The “Feeding” study was approved by the Ethical Committees of the “Ospedale Pediatrico Bambino Gesù” (OPBG) and the SCH. The study was in accordance with the national and international regulations, and the Declaration of Helsinki (2000). All pregnant women who participated in this study signed an informed consent.

### 2.2. Anthropometrics and Clinical Evaluation

The women’s anthropometrics were estimated [12] at baseline and at each outpatient visit during pregnancy until childbirth. The body mass index (BMI) was calculated as kg/m^2^, and the obesity-status-classified [13] GDM was diagnosed as according to the American Diabetes Association’s Standards of Care [14].

We calculated the gestational weight gain (GWG) as the difference between the pre-pregnancy weight to the weight at the time of delivery, and defined adequate GWG in relation to pre-pregnancy BMI (12.5–18.0 kg in underweight; 11.5–16.0 kg in normal weight; 7.0–11.5 kg in overweight, and 5.0–9.0 kg in obesity, respectively), or as inadequate or excessive if the weight gain was below or exceeded values recommended for pre-pregnancy BMI classes, respectively.

We collected sociodemographic (race; level of education; profession; smoking; and parity) and anthropometric variables of both parents. Finally, we estimated the newborns’ with the following variables: body weight (BW); body length (BL); and head circumference (HC) at birth. Standard deviation scores (SDS) for infant weight and height were calculated following the Italian Neonatal Study Chart [15].

### 2.3. Samples Collection and Cytokines Analysis

Fasting blood samples were withdrawn from all the participants at week ~38. Red blood cells were isolated from the serum within 2 h, and erythrocyte membranes were isolated using a standard procedure. Briefly, after plasma centrifugation (980 rpm, 18 min), RBCs were added with acid citrate dextrose, washed with distilled water (10:1), centrifuged (4000 rpm, 5 min) four times, and then were frozen at −80 °C and immediately stored.

Serum was stored for the later assay of the following adipokines and cytokines: human total adiponectin/Acrp30 Quantikine ELISA (R&D Systems, Inc. Catalog# DRP300, Minneapolis, MN, USA); leptin (Quantikine™ ELISA Human Leptin Immunoassay, Catalog # DLP00; detection range 15.6–1000 pg/mL; sensitivity 7.8 pg/m); sCD14 (RayBio^®^ Peachtree Corners, GA, USA, Human soluble Cluster of Differentiation 14 ELISA Kit, Catalog #: ELH-CD14; detection range 6–6000 pg/mL; sensitivity 6 pg/mL); sTNFR-II (RayBio^®^ Human soluble Tumor necrosis factor receptor II ELISA Kit, Catalog #: ELH-TNFR2; detection range 5–2000 pg/mL; sensitivity 5 pg/mL); IL-6 (RayBio^®^ Human IL-6 ELISA Kit, Catalog #: ELH-IL6; detection range 3–1000 pg/mL; sensitivity 3 pg/mL); IL-8 (RayBio^®^ Human IL-8 ELISA Kit, Catalog #: ELH-IL8; detection range 1–600 pg/mL; sensitivity 1 pg/mL); IL-10 (RayBio^®^ Human IL-10 ELISA Kit, Catalog #: ELH-IL10; detection range 1–150 pg/mL; sensitivity 1 pg/mL); MCP-1 (RayBio^®^ Human monocyte chemoattractant protein 1 ELISA Kit, Catalog # ELH-MCP1; detection range 2–500 pg/mL; sensitivity 2 pg/mL); endotoxin (LAL Chromogenic Endpoint Assay, Hycult Biotech, PB Uden, The Netherlands, Catalog # HIT302; minimum detection limit of 0.04 EU/mL; concentration range 0.04–10.0 EU/mL); and ICAM-1 (Catalog No: MBS2515841 MYBIOSOURCE, San Diego, CA, USA; detection range 0.31–20 ng/mL; 0.19 ng/mL).

### 2.4. Lipid Extraction from RBC Membranes and Gas Chromatography Analysis of Fatty Acid Methyl Esters (FAME)

RBC membranes were isolated using the standard procedure [16] and as described earlier [9,10]. The supernatant containing membranes was transferred in gas-chromatography vials and the solvent was removed. Methyl-C11 (2 mg/mL) was added as an internal standard. FAME analysis was made by using a fast gas-chromatography/flame-ionization detector (2010 Plus with an autosampler AOC-20i, Shimadzu, Kyoto, Japan) as described in detail [9,10]. Chromatograms were integrated and identified by comparing the retention times and the peak area against those of a commercial lipid standard containing 52 fatty acids (GLC 463, Elysian, MN, USA) and a conjugated linoleic acids mixture (UC-59M Nuchek; Elysian, MN, USA). Quantitative data were obtained through interpolation of the relative areas vs. the internal standard (Methyl-C11) area. Data are shown as FAME concentration (ng/mL) or percentage (% of total FAME), respectively.

### 2.5. Statistical Analysis

Descriptive analysis was conducted for categorical data through frequencies and percentages, for continuous data through mean and standard deviation (SD), or median and inter-quartile range (IRQ) as appropriate. A correlation plot was used to evaluate the correlation between FAs and cytokines (R corplot package version 0.92).

As cytokines were found to be not normally distributed (Shapiro–Wilk normality test), we analyzed their association to FAs with quantile regression for the 50th quantile (median), a method that is applied when the conditions of linear regression are not satisfied and that is robust to outliers.

A quantile regression model for each cytokine and FAs one by one at the time was fitted while adjusting for confounders. We considered as confounders variables: maternal age, smoking, GDM, hypertension, gestational age, gestational weight gain, and maternal weight at week 38.

For each cytokine, a multiple quantile regression model was fitted by including each FA associated with the cytokine with a *p*-value < 0.1 of the previous models and the above confounders. Finally, we assessed, in a quantile regression mediation model, three distinct clinical variables (insulin, pre-pregnancy BMI, and GWG), to explore their possible mediating roles in the association between FAs and cytokines, resulting in significant associations in the univariate quantile regressions.

The same approach was used to explore the association of patterns between the FAs and the cytokines, i.e., TFAs (including C16:1 t1, C18:1 t9, and C18:1 t11, respectively), SFAs (including C12:0, C14:0; C15:0, C16:0, C17:0, C18:0, C19:0, C20:0, and C22:0, C24:0. n-3, respectively), PUFAs (including C20:3, C20:4, C20:5, C22:5, and C22:6, n-6, respectively), and PUFAs (including C20:3, C20:4, and C22:2, respectively).

Mediation analysis was performed with mediation R package version 4.5.0, and quantile regression with the quantreg R package version 5.94. Heatmap plot was performed with theheatmap.2 function from gplots package version 3.1.3.

## 3. Results

### 3.1. Distribution of RBC Fatty Acids and Circulating Cytokines

Il-6 was assessed in 390 women from the whole cohort of 847 women. IL-8 was evaluated in 496 women, while all the other cytokines were examined in 525 women. To keep the relevant IL-6 and IL-8 data in our analysis, we only enclosed the data of 250 women who had complete datasets of FAs and cytokines. No significant differences were found in the lipid RBC membrane profile and cytokine milieu between women with complete and incomplete datasets (see Appendix A). Table 1 summarizes their anthropometrics and demographics. Owing to the skewed distribution of the cytokines, we reported the median values and quartiles of each circulating cytokine in Table 2.

We reported the mean values and quartiles of the FAs in Appendix A.

### 3.2. Quantile Regression between RBC Fatty Acids and Cytokines

Several statistically significant associations between FAs and cytokines were found, as reported in Table 3. No significant association was found between any FA and values of insulin, IL-6, IL-10, and endotoxin.

Figure 1 shows their correlation plots (Appendix A shows the complete correlation plot.) No significant associations were found between any FA and values of insulin, IL-6, IL-10, and endotoxin. Figure 2 shows a heatmap of the scaled coefficients of the quantile regressions for the significant relationships.

Figure 2 shows a heatmap of the coefficients of each regression scaled.

### 3.3. Multiple Quantile Regression between RBC Fatty Acids and Cytokines

Multiple quantile regression was used to model relationships between each cytokine and the FAs that resulted in significant associations in the quantile regression reported above with *p* < 0.1 (C12:0, C14:0, C15:0, C16:0, C18:0, C24:0, C12:1, C22:6, C20:1c11, C20:0, C18:2; C18:1 c13/c14, C22:3/C22:4; C24:1, C22:2, C22:5, C19:1, C20:1; C18:1 c9, C22:0, and C14:1, respectively), while adjusting for confounding (including maternal age, smoking, GDM, hypertension, GWG, and maternal weight at week 38 of pregnancy). Final models are reported in Table 4, which also shows the results of multiple quantile regression for each cytokine considering MUFAs, SFAs, and PUFAs, jointly, and with confounders.

### 3.4. Mediation Analysis

Conducting mediation analysis is beneficial to identify the mechanisms behind an observed relationship between an independent variable and a dependent variable through the inclusion of a third variable, known as a mediator. Rather than a direct relationship between the independent and the dependent variable, a mediation model proposes that the independent variable influences the mediator, which in turn influences the dependent variable, as displayed in Figure 3.

We evaluated the three distinct variables, namely insulin, pre-pregnancy BMI, and GWG in a quantile regression mediation model to investigate their possible mediating roles in the regulation of the inflammatory patterns for all the pairwise combinations of FA and cytokines that resulted in significant associations in the previous models. Despite several pairwise effects between FAs, cytokines, and clinical variables being found significant, the mediation analysis revealed no relevant mediating roles of the clinical variables in the inflammatory patterns, suggesting that the pairwise effect between fatty acids and cytokines plays a predominant role in the inflammatory effects (Table 5).

## 4. Discussion

The findings we obtained in this study demonstrate that dietary FAs contribute to maintaining the balance between pro-inflammatory and anti-inflammatory responses during pregnancy. In doing so, they favor healthy pregnancy outcomes. It is worth noting that associations of single dietary FAs and circulating cytokines were found to be independent of the obesity status, of the amount of weight gain during pregnancy, and of the insulin levels as the mediation analysis demonstrated. Indeed, differently from what we expected, there was no mediation of body weight, weight gain, or insulin levels found in the association between fatty acids and inflammation. FAs, with regard to SFAs and n-3 PUFAs, are known to influence both sides of insulin metabolism, sensitivity, and glucose-induced insulin secretion [5,6]. Nevertheless, in our series of pregnant women, there was no significant association found between any FA and circulating insulin.

It is also important to highlight that in our analysis we looked at the association with inflammatory molecules of single FAs but also of groups of FAs, namely SFAs, PUFAs, and TFAs. We selected for investigation a few molecules that affect different paths of inflammation and the immune response, and whose circulating levels have been associated with pregnancy and fetus-related health outcomes [1,2]. At multivariable quantile regression, we found s*CD14*, *TNF-RII,* and *ICAM-1* to be negatively associated with SFAs. *TNF-RII* was also found to be negatively associated with MUFA levels. No association emerged with PUFAs. Furthermore, several adipokines and cytokines, i.e., adiponectin, were associated with single FAs, but indexes of adiposity were the best predictors for others such as leptin.

### 4.1. Adipokines: Leptin and Adiponectin

First, we focused on leptin and adiponectin, whose circulating levels are influenced by the fat mass that is secreted by the adipose tissue. Circulating leptin increases throughout pregnancy, with the highest levels observed in the third trimester owing to the greater body fat stores. Conversely, adiponectin levels decrease [17,18].

Leptin plays an important role in regulating the energy balance and metabolism, including insulin sensitivity. It affects several physiological processes that are important for pregnancy, such as placental development and function, fetal growth, and development through promoting angiogenesis, and by stimulating the release of insulin-like growth factor-1. Leptin also helps to regulate placental function by increasing the uptake of glucose and amino acids by the placenta [19]. In our cohort, quantile regression demonstrated leptin values to be associated with the maternal body weight at the end of pregnancy.

Adiponectin is thought to play a role in maintaining healthy pregnancy outcomes by regulating insulin sensitivity, maternal glucose and lipid metabolism, reducing inflammation, and promoting proper placental function, fetal growth, and development [20]. Low levels of adiponectin have been associated with pregnancy complications such as GDM [21]. In our series, we found an inverse association of adiponectin levels with mixtures of C22:3 and C22:4, and a positive association with C18:1 c13/c14.

### 4.2. Inflammatory and Anti-Inflammatory Cytokines

*sCD14* is a protein that is released into circulation from immune cells, including monocytes and macrophages, and plays an important role in the immune response, as well as in inflammation and metabolism. Its levels have been found elevated in women with pre-eclampsia and in those who deliver preterm [2]. sCD14 has been shown to inhibit the action of adiponectin [22], and since both sCD14 and adiponectin tended to be positively associated to levels of C22:6, we speculate that this FA might play a role in the anti- and the pro-inflammatory balance during pregnancy, also contributing, in this way to brain development. C22:6 (or DHA) is the key FA for brain development [23].

Endotoxin is a component of the outer membrane of Gram-negative bacteria and the most powerful activator of the immune system. This molecule crosses the placenta and activates the fetal immune system. Endotoxin exposure during pregnancy has been associated with preterm birth, pre-eclampsia, and fetal growth restriction [2]. Endotoxin was found to be positively associated with the concentration of C20:1, a FA that is found in plant oil and nuts and was found to be negatively associated to the behenic acid C22:0 (obtained from rapeseed oil and peanuts).

There was also a trend for a positive association found between the nervonic-acid C24:1 and levels of both IL-6 and IL-10. Nervonic acid is among the group of cerebrosides which account for approximately 40% of the total FAs in sphingolipids [24]. These associations involving anti- and pro-inflammatory molecules on one side, and FAs that are key players in neurodevelopment being constituents of neuronal cell membranes on the other side, recall the role of a balanced inflammation on neurogenesis and plasticity, and suggest another way of FAs to contribute to the healthy neurodevelopment. Indeed, excessive inflammation would impair fetal brain development [25]. IL-6 is an important regulator of immune function during pregnancy. It helps to balance the maternal immune system, preventing rejection of the fetus while also protecting the mother against infections. The molecule is involved in the invasion of trophoblast cells that will set up the placenta; and helps to regulate placental development. At the end of pregnancy, its role is fundamental for the onset of labor. Nevertheless, IL-6 is involved in the development of the fetal brain promoting neuronal proliferation, differentiation, and survival [26].

IL-10 is an anti-inflammatory cytokine that is generated by the placenta, and it helps to maintain a balance between the maternal immune system and the developing fetus. IL-10 levels are elevated during pregnancy, particularly in the third trimester [27].

IL-8 acts as a chemoattractant to white blood cells, particularly neutrophils, to the site of infection or inflammation. IL-8 is produced by the placenta and helps to regulate the growth and development of blood vessels in the placenta. Levels can be elevated during pregnancy, particularly in cases of preterm labor and delivery [28]. In our series, these values were found to be associated with maternal weight at week 38.

MCP-1 concentrations were found to be positively associated with C16:0 levels, a SFA which is known to worsen insulin sensitivity and potentiate glucose induced insulin secretion, hence favoring hyperinsulinemia [6]. MCP-1 is a chemokine produced by a variety of cell types, including endothelial cells, macrophages, and smooth muscle cells, and it functions to recruit monocytes and other immune cells to sites of inflammation; and plays a role in trophoblast invasion and migration [29].

While no associations were found between FAs and levels of *TNFR-II*, significantly elevated levels of the latter molecule were observed in association to GDM. sTNF-R acts as a decoy receptor for TNF. It binds to TNF and prevents it from binding to its cell surface receptors, which can reduce the level of inflammation in the body. There are two types of sTNF-R: sTNF-RI and sTNF-RII, which have different affinities for TNF. During pregnancy, the levels of sTNF-R can be affected. TNF-alpha is involved in several processes during pregnancy, including implantation, placental development, and maintenance of pregnancy [2]. However, excessive production of TNF-alpha can have detrimental effects on pregnancy, including preterm labor, pre-eclampsia, and fetal growth restriction. Studies have shown that sTNF-R levels are increased during pregnancy, and that they may play a role in protecting the fetus from the effects of excessive TNF-alpha [30].

The only significant association of ICAM-1 at the multivariate quantile model observed was inverse with levels of the SFA myristic acid. ICAM-1 has been implicated in the adhesion and migration of trophoblast cells and in the regulation of angiogenesis. Increased levels of ICAM-1 have been found in the placentas of women with pre-eclampsia, and the molecule has been implicated in the pathogenesis of fetal growth restriction [2].

Findings of our study confirm that there may be complex interactions between different biomarkers during pregnancy, and that changes in the FA levels have downstream effects on the levels of cytokines and other biomarkers of inflammation in pregnant women from a general population on a free diet and no n-3 PUFA supplementation.

Studies in the literature mostly have investigated beneficial effects of n-3 PUFAs supplementation during pregnancy on maternal and neonatal outcomes [31]. Supplementation of overweight/obese pregnant women with dietary n-3 PUFAs for >25 weeks has been found to reduce inflammation in maternal adipose and placental tissues [32]. We demonstrated that even the intake of single FAs belonging to certain patterns can have beneficial or detrimental effects in terms of inflammatory balance during pregnancy.

Strengths of our study include the profiling of fatty acids on the erythrocyte membranes as markers of dietary intake of lipids during pregnancy instead of referring to plasma levels or to dietary recall. Furthermore, we investigated a population that was a general one on a free diet. Nevertheless, several of the associations that we have described are of an uncertain significance and, owing to the cross-sectional design of study, the causative direction of these relationships cannot be established. Longitudinal studies of healthy and unhealthy women tracking the dietary intake of fats and levels of cytokines along the whole pregnancy are warranted to better understand the effects of dietary FAs on the mother’s health and on the pregnancy outcomes, including the time of delivery. Indeed, we limited our investigation to healthy women, and owing to economical restrains we assessed FAs and inflammatory molecules at the end of the pregnancy. By study design, we excluded women who gave birth before 38 weeks of gestation from the study, although it would have also been of interest to investigate these associations in such groups too. Reduced levels of n-3 PUFAs [33] and increased inflammation have been associated with premature parturition [2].

## 5. Conclusions

Our findings demonstrate the importance of maternal diet during pregnancy to maintain maternal metabolic profile and fetus sustainability. These findings are important when designing dietary strategies to optimize maternal metabolism during pregnancy for a successful pregnancy outcome. The study provides important insights into the complex interplay between different biomarkers during pregnancy and highlights the need for a holistic approach to studying maternal health. Further research is needed to fully understand the implication of these associations, and to explore the potential mechanisms underlying the observed effects.

## Figures and Tables

**Figure 1 nutrients-15-02432-f001:**
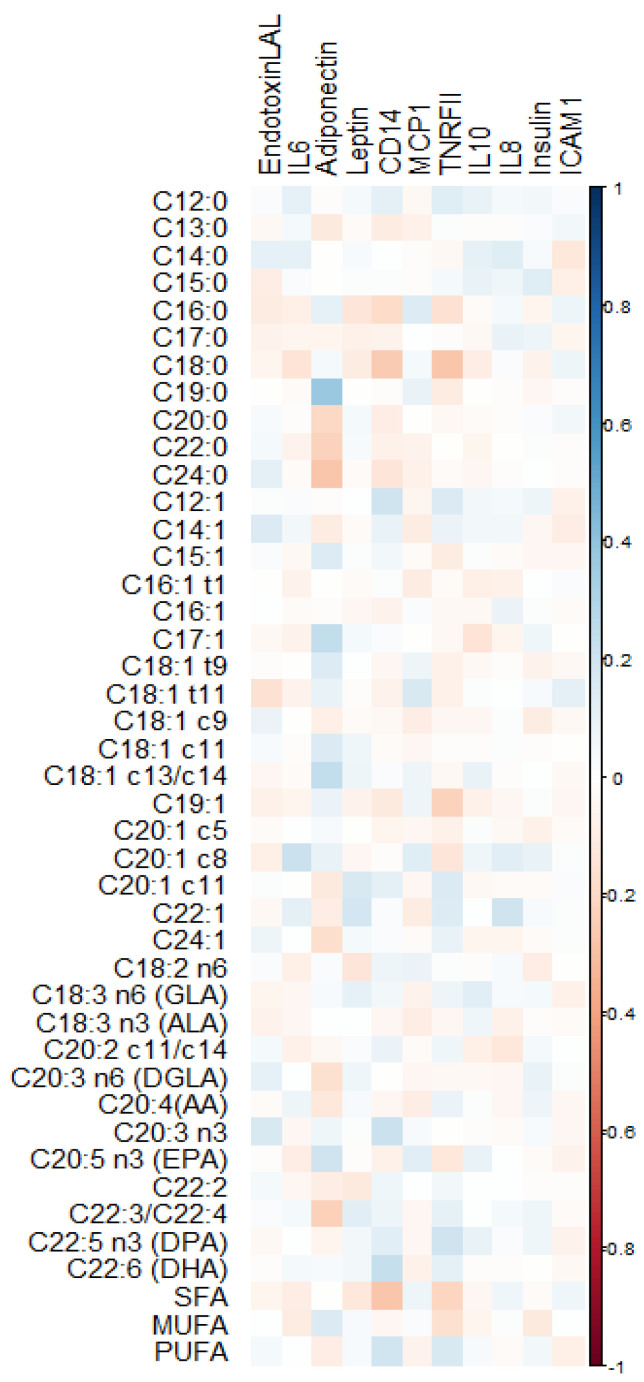
Correlation plot of RBC membrane fatty acids (%) and circulating cytokines at 38 weeks of pregnancy. All fatty acids had cis conformation except for those with “t”, which are trans-isomers. To improve readiness, we have added for the main polyunsaturated fatty acids the location of the double bond three or six carbons (n3 or n6) away from the methyl carbon and common name. LA, linoleic; GLA, gamma-linolenic; ALA, alpha-linolenic; AA, arachidonic; DGLA, dihomo-γ-linolenic; EPA, eicosapentaenoic; DPA, docosapentaenoic; DHA, docosahexaenoic or cervonic acid, endotoxin (EU/mL); adiponectin (mcg/mL); leptin (ng/mL); soluble cluster of differentiation 14 (sCD14) (mcg/mL); tumor necrosis factor receptor II, TNFR-II (pg/mL); interleukin–6, IL–6 (pg/mL); interleukin–8, IL–8 (pg/mL); interleukin–10, IL–10 (pg/mL); insulin (μUI/mL); intercellular adhesion molecule 1, ICAM–1 (pg/mL); and monocyte chemoattractant protein 1, MCP–1 (ng/mL).

**Figure 2 nutrients-15-02432-f002:**
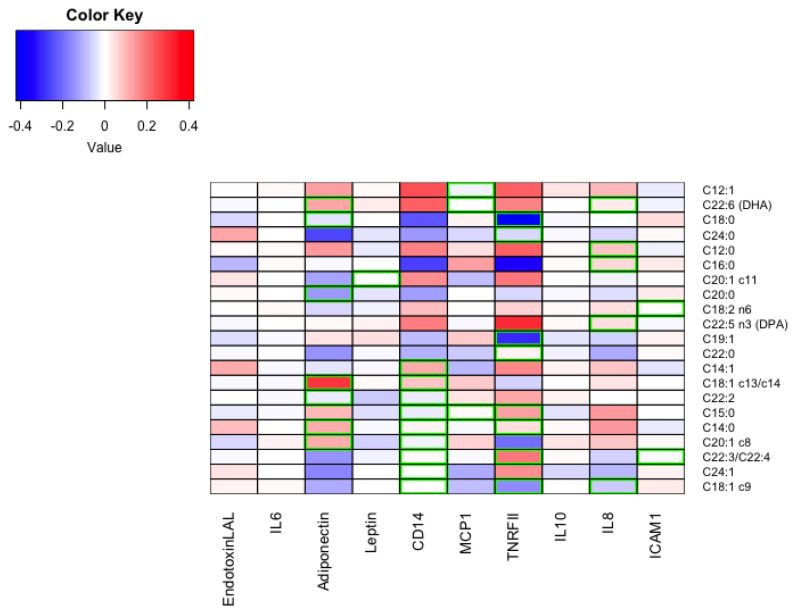
Heatmap of the scaled coefficients of the univariate quantile regressions for the significant relationships. DPA, docosapentaenoic; DHA, docosahexaenoic or cervonic acid, endotoxin (EU/mL); adiponectin (mcg/mL); leptin (ng/mL); soluble cluster of differentiation 14 (sCD14) (mcg/mL); tumor necrosis factor receptor II, TNFR–II (pg/mL); interleukin–6, IL–6 (pg/mL); interleukin–8, IL–8 (pg/mL); interleukin–10, IL–10 (pg/mL); insulin (μUI/mL); intercellular adhesion molecule 1 (ICAM–1) (pg/mL); monocyte chemoattractant protein 1, MCP–1 (ng/mL).

**Figure 3 nutrients-15-02432-f003:**
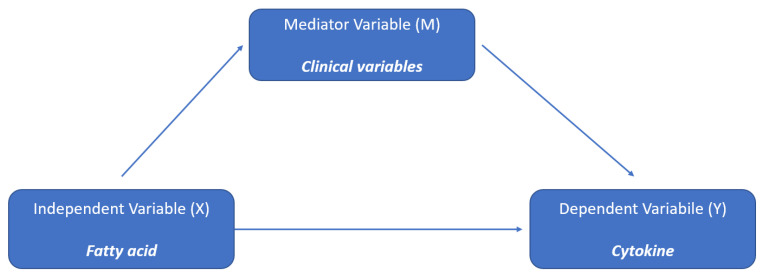
Schematic representation of the mediation analysis.

**Table 1 nutrients-15-02432-t001:** Anthropometrics and demographics of pregnant women. Data are shown as mean and standard deviation (SD) or number and percentage (%). GWG: gestational weight gain.

Mothers (*n* = 250)	
Age (years)	32.6 (29.0–37.0)
Maternal smoking	37 (10.8%)
Gestational diabetes	9 (3.6%)
Body mass index and nutritional status	
Pre–pregnancy BMI (kg/m^2^)	23.0 (19.7–24.8)
Underweight	20 (8%)
Normal weight	169 (67.6%)
Overweight	40 (16%)
Obese	21 (8.4%)
	Education
Primary school	5 (2%)
Secondary school	37 (14.8%)
High school	117 (46.8%)
Bachelor’s degree	91 (36.4%)
Gestational weight gain	
GWG (kg)	13.37 (10.0–17.00)
Inadequate	67 (26.8%)
Adequate	91 (36.4%)
Excessive	92 (36.8%)
Delivery	
Vaginal	181 (72.4%)
Cesarean	69 (27.6)
Gestational age	39.1 (38–40)
Newborn birth weight (g)	3383 (3060–3670)
Newborn birth length (cm)	50.6 (49–52)

**Table 2 nutrients-15-02432-t002:** Median values and IQRs of circulating cytokines and insulin at 38 weeks of pregnancy.

Cytokines	
Adiponectin (mcg/mL)	4.92 (2.39–9)
Leptin (ng/mL)	25.05 (12.76–43.54)
Endotoxin (EU/mL)	2.35 (1.85–3.13)
Soluble cluster of differentiation 14 (sCD14) (mcg/mL)	1.84 (1.14–3.06)
Tumor necrosis factor receptor II (TNFR-II) (pg/mL)	797.6 (531.0–1010.0)
Interleukin-6 (IL-6) (pg/mL)	24.09 (7.69–64.61)
Interleukin-8 (IL-8) (pg/mL)	6.94 (3.39–12.66)
Interleukin-10 (IL-10) (pg/mL)	19.59 (10.26–35.38)
Insulin (μUI/mL)	16.2 (8.3–22)
Intercellular Adhesion Molecule 1 (ICAM-1) (pg/mL)	354.55 (226.82–662.15)
Monocyte Chemoattractant Protein 1 (MCP-1) (ng/mL)	26.19(16.23–36.96)

**Table 3 nutrients-15-02432-t003:** Quantile regression models for each cytokine vs. each FAs adjusted for confounders. Only statistically significant associations are shown.

FA (%)	Coefficient	*p*-Value
Adiponectin (mcg/mL)		
C12:0	0.82	0.020
C20:0	−5.16	0.028
C24:0	1.20	<0.001
C18:1 c13/c14	1.50	0.015
C20:1 c11	−4.89	0.037
C24:1	−0.52	0.037
C22:3/C22:4	−1.07	0.002
Leptin (ng/mL)		
C22:1	−6.90	0.040
sCD14 (mcg/mL)		
C12:0	0.25	0.009
C16:0	−0.12	0.010
C18:0	−0.15	0.001
C20:0	−1.20	0.023
C24:0	−0.16	0.008
C12:1	0.43	<0.001
C20:1 c11	1.50	0.020
C18:2 n6	0.07	0.040
C22:6 (DHA)	0.23	<0.001
SFA	−0.06	0.001
TNRF-II (pg/mL)		
C16:0	−31.50	0.010
C18:0	−55.80	<0.001
C12:1	81.70	0.003
C19:1	−517.8	0.002
C20:1 c11	−385.60	0.040
C22:3/C22:4	−70.08	0.040
C22:5 n3 (DPA)	74.82	0.040
SFAs	−14.77	0.010
MUFAs	−13.36	<0.001
PUFAs	12.81	0.006
ICAM-1 (pg/mL)		
C18:0	−20.14	0.002
C14:1	−52.69	<0.001
SFAs	8.09	0.010
IL-8 (pg/mL)		
C14:0	2.49	0.002
C15:0	3.02	0.040
C22:0	−2.16	0.010
C12:1	1.11	0.020
C24:1	−0.5	0.015

**Table 4 nutrients-15-02432-t004:** Multiple quantile regression models for each cytokine vs. FAs adjusted for confounders. Only statistically significant associations are shown.

FAs (%)	Coefficient	*p*-Value
Adiponectin (mcg/mL)		
C18:1 c13/c14	1.20	0.020
C22:3/C22:4	−1.44	0.008
C22:6 (DHA)	0.62	0.060
Endotoxin (EU/mL)		
C20:1 c5	−0.90	0.030
C22:1	−0.40	0.050
Leptin (ng/mL)		
Maternal body weight at week 38	0.90	<0.001
sCD14 (mcg/mL)		
C22:6 (DHA)	0.20	0.060
TNRF-II (pg/mL)		
C24:1	47.90	0.070
GDM	523.1	0.006
ICAM-1 (pg/mL)		
C14:0	−86.80	0.045
C14:1	−63.30	0.060
GDM	688	0.060
Smoking habit	133	0.090
IL-6 (pg/mL)		
C24:1	3.70	0.10
IL-8 (pg/mL)		
C19:1	−5.80	0.090
C18:2 n6	0.50	0.060
Maternal age	0.10	0.040
IL-10 (pg/mL)		
C24:1	−2.50	0.060
MCP-1 (ng/mL)		
C15:1	0.80	0.040
C18:1 t11	−9.50	0.070
Maternal age	0.40	0.090
Association of patterns of FAs with cytokines		
sCD14 (mcg/mL)		
SFAs	−0.07	0.003
ICAM-1 (pg/mL)		
SFAs	7.77	0.080
TNRF-II (pg/mL)		
SFAs	−12.1	0.050
MUFAs	−17	0.003

**Table 5 nutrients-15-02432-t005:** Statistically significant associations between mediators and FAs.

Cytokine (y)	Fatty acid (x)	Mediator (m)	y~x Pval	m~x Pval	Mediation Pval
IL-8	C14:0	Insulin	0.018	0.034	0.408
TNFR-II	C22:5	Insulin	0.040	0.009	0.548
sCD14	C16:0	Pre-gravid BMI	0.012	0.015	0.932

## Data Availability

The data presented in this study are available on request from the corresponding author. The data are not publicly available due to privacy and ethical restrictions.

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
