# Peer review of "Dietary Fatty Acids Contribute to Maintaining the Balance between Pro-Inflammatory and Anti-Inflammatory Responses during Pregnancy"

_nutrients, 2023, doi:10.3390/nu15112432_

Round 1
Reviewer 1 Report
Introduction:
Reference is needed ''n parallel the woman's body experiences metabolic changes to accommodate the increased metabolic demands of fetal growth and development.''
In last paragraph of itroduction:Justification is required to support the investigation of the association at 38weeks of pregnancy. Also to specify these molecules which are suggested as important for immune function, inflammation and metabolism
End of Discussion: it would add clarify if further thoughts and recommendations are provided in relation to appropriate studies to investigate the causative direction of these association.
Author Response
Introduction:
Reference is needed ''n parallel the woman's body experiences metabolic changes to accommodate the increased metabolic demands of fetal growth and development.''
We quoted ref. 3 that is an extensive review covering the whole paragraph.
In last paragraph of introduction: Justification is required to support the investigation of the association at 38weeks of pregnancy. Also to specify these molecules which are suggested as important for immune function, inflammation and metabolism.
Done. From the text: “…The lipid profile on RBC membranes reflected quality of dietary fats in the previous 40 days. Cytokines were tested at 38 weeks when their levels were expected to change favoring the labor onset.
For these purposes, … We selected for testing molecules reflecting different facets of the inflammation. List included adiponectin, leptin, soluble Cluster of Differentiation 14 (sCD14), soluble Tumor necrosis factor receptor II (sTNFR-II), IL-6, IL-8, IL-10, monocyte chemoattractant protein 1 (MCP-1), endotoxin, and human Intercellular Adhesion Molecule 1 (ICAM-1).”.
End of Discussion: it would add clarify if further thoughts and recommendations are provided in relation to appropriate studies to investigate the causative direction of these association.
Done. From the text “…Longitudinal studies of healthy and unhealthy women tracking dietary intake of fats and levels of cytokines along the whole pregnancy are warranted to understand deeper the effects of dietary FAs on the mother’s health and on the pregnancy outcomes, including time of delivery. Indeed, we limited our investigation to healthy women and owing to economical restrains we tested FAs and inflammatory molecules at the end of the pregnancy. By study design, we excluded from the study women who gave birth before 38 weeks of gestation while it would be of interest to investigate these associations in such group too. Reduced levels of n-3 PUFAs [33] and increased inflammation have been associated with premature parturition [2].”
Reviewer 2 Report
This is an interesting cross-sectional study relating fatty acid measurements in erythrocytes to (simultaneous) measurements of biomarkers reflecting inflammation status. The blood was drawn from fasting pregnant women shortly (12h-24h) before they delivered their child. The biochemical / lab work seems to be well conducted. The report brings results from statistical analyses relating a large number of different fatty acids to a long list of inflammation biomarkers; although this seems like a rather unfocused approach, the report would be of interest to researchers in the field.
I have the following comments to the authors:
The authors state (in the beginning of section 3.1) that “From the whole cohort of 847 women, we enclosed data of 250 women who had complete dataset”. Information is needed about how this narrowing of the initial group of 847 women down to 250 women affected the characteristics of the study group. A flow diagram would be helpful.
Since the levels of the measured biomarkers may vary with time at which they were assessed during pregnancy, information is needed about distribution of gestational age at blood drawing (which will be nearly the same as gestational age at delivery) of these 250 women.
Information about birth weight and birth length would also be helpful.
(No comments)
Author Response
This is an interesting cross-sectional study relating fatty acid measurements in erythrocytes to (simultaneous) measurements of biomarkers reflecting inflammation status. The blood was drawn from fasting pregnant women shortly (12h-24h) before they delivered their child. The biochemical / lab work seems to be well conducted. The report brings results from statistical analyses relating a large number of different fatty acids to a long list of inflammation biomarkers; although this seems like a rather unfocused approach, the report would be of interest to researchers in the field.
We thank the reviewer for the comment, and we specified that these molecules reflect different facets of the inflammation.
I have the following comments to the authors:
The authors state (in the beginning of section 3.1) that “From the whole cohort of 847 women, we enclosed data of 250 women who had complete dataset”. Information is needed about how this narrowing of the initial group of 847 women down to 250 women affected the characteristics of the study group. A flow diagram would be helpful.
We opted to insert in results a clear description instead of a flow chart (too many figures and tables already in the manuscript. From the text: “..Il-6 was tested in 390 women from the whole cohort of 847 women. IL-8 was tested in 496 women, while all the other cytokines were tested in 525 women. To keep in the analysis IL-6 and IL-8 we enclosed only data of 250 women who had complete dataset of FAs and cytokines. No significant differences were found in lipid RBC membrane profile and cytokine milieu between women with complete and uncomplete dataset (see supplementary Figure 1)…”
Since the levels of the measured biomarkers may vary with time at which they were assessed during pregnancy, information is needed about distribution of gestational age at blood drawing (which will be nearly the same as gestational age at delivery) of these 250 women.
Right. We specified gestational age in Table 1 and in the text that the blood test was done around week 38. Any way gestational age was considered in all the multivariable regression models.
Information about birth weight and birth length would also be helpful.
Added in Table 1.